

# Chlorine partitioning in the lowermost Arctic vortex during the cold winter 2015/2016

Andreas Marsing[1,2], Tina Jurkat-Witschas[1], Jens-Uwe Grooß[3], Stefan Kaufmann[1], Romy Heller[1], Andreas Engel[4], Peter Hoor[2], Jens Krause[2,a], and Christiane Voigt[1,2]

[1]Deutsches Zentrum für Luft- und Raumfahrt, Institut für Physik der Atmosphäre, Oberpfaffenhofen, Germany
[2]Johannes-Gutenberg-Universität Mainz, Institut für Physik der Atmosphäre, Mainz, Germany
[3]Forschungszentrum Jülich, Institut für Energie- und Klimaforschung – Stratosphäre (IEK-7), Jülich, Germany
[4]Goethe Universität Frankfurt, Institut für Atmosphäre und Umwelt, Frankfurt, Germany
[a]now at: Excelitas Technologies GmbH & Co. KG, Wiesbaden, Germany

**Correspondence:** Andreas Marsing (andreas.marsing@dlr.de)

**Abstract.** Activated chlorine compounds in the polar winter stratosphere drive catalytic cycles that process ozone and methane, whose abundances are highly relevant to the evolution of global climate. The present work introduces a novel dataset of in situ measurements of relevant chlorine species in the Arctic lowermost stratosphere from the aircraft mission POLSTRACC/GW-LCYCLE/SALSA during winter 2015/2016. The major stages of chemical evolution of the lower polar vortex are presented in a consistent series of high resolution mass spectrometric observations of HCl and $ClONO_2$. Simultaneous measurements of CFC-12 are used to derive total inorganic chlorine ($Cl_y$) and active chlorine ($ClO_x$). The new data highlight an altitude dependent shift in the pathway of chlorine deactivation through the recovery of the reservoir species from $ClONO_2$ to HCl in the lowermost vortex below the 380 K isentropic surface. Further, we show that the Chemical Lagrangian Model of the Stratosphere (CLaMS) is generally able to reproduce the chemical evolution of the lower polar vortex chlorine budget, except from a bias in HCl concentrations. The model is used to relate local measurements to the vortex-wide evolution. The results are aimed at fostering our understanding of the climate impact of chlorine chemistry, providing new observational data to complement satellite data and assess model performance in the climate sensitive upper troposphere and lower stratosphere region.

## 1 Introduction

Understanding the processes that affect the abundance of climate relevant gases in the Arctic polar upper troposphere and lower stratosphere (UTLS) is crucial for a reliable estimation of their current and future impact on the radiation budget and thus on global climate. One of the principal mechanisms for the depletion of ozone ($O_3$) in the polar stratosphere is a photochemically driven catalytic cycle where gas phase molecular chlorine and chlorine oxides provide the catalyst to reduce ozone molecules to oxygen (Molina and Molina, 1987). All of the chlorine species that contribute to this reaction are summarised under the term "active chlorine", which is abbreviated with $ClO_x$ and comprises Cl, $Cl_2$, ClO, ClOOCl, OClO and HOCl. Conversely, the passive molecules hydrogen chloride (HCl) and chlorine nitrate ($ClONO_2$, recent review by von Clarmann and Johansson,



2018) contribute only indirectly to ozone depletion and are therefore called reservoir species. Besides chlorine, also bromine compounds are important. Bromine has the second most prominent role in ozone depletion cycles, with a higher reaction efficiency but far lower abundance, compared to chlorine (e.g. Daniel et al., 1999).

Chlorine is introduced into the stratosphere via chloroflourocarbons (CFCs) from anthropogenic sources. While the emission
of CFCs has nearly ceased as a consequence of the Montreal Protocol and its amendments, Montzka et al. (2018) report a slow down of the decrease rate of atmospheric CFC-11 since 2012 at some measurement sites, indicating unexpected new sources. In addition, chlorinated very short-lived substances (VSLS) appear to contribute an increasing fraction to the stratospheric chlorine burden (Hossaini et al., 2019). Therefore, continued monitoring of chlorine species and their predecessors is a crucial necessity for projections of the future climate, as the reservoir species remain in the stratosphere for decades due to their long
chemical lifetimes (WMO, 2011) and chlorine-catalysed ozone depletion will continue to occur throughout the 21st century.

Polar stratospheric clouds (PSCs) play a crucial role in ozone depletion, as they provide a surface for the heterogeneous reactions that produce active chlorine from the reservoir species (Crutzen and Arnold, 1986; Drdla and Müller, 2012). As these solid condensates of nitric acid-trihydrate (Voigt et al., 2000a; Fahey et al., 2001), water ice (Toon et al., 2000) or liquid ternary solution particles (Carslaw et al., 1994; Voigt et al., 2000b) occur only at very low temperatures below about 196 K,
they can exist inside the stratospheric polar vortices. During winter radiative cooling over the poles leads to the formation of the polar vortex which acts as a transport barrier between polar and mid-latitude stratospheric air. As sunlight returns towards spring, photochemical reactions are initiated that lead to chlorine activation and to the formation of the ozone hole. It is less pronounced in the Arctic than in the Antarctic due to a less stable polar vortex as a consequence of enhanced planetary wave activity on the northern hemisphere (e.g. Solomon et al., 2014; Matthias et al., 2016).

A common definition locates the lowermost stratosphere (LMS) between the tropopause and the 380 K isentrope (Holton et al., 1995). Although it shares most features with the greater part of the entire stratosphere, the vicinity of the tropopause makes it prone to stratosphere-troposphere exchange (STE). Also, in the polar winter LMS, the transport barrier of the polar vortex fades towards the tropopause, enabling isentropic transport and mixing (e.g. Werner et al., 2010) with air from the lower latitude stratosphere and the extratropical transition layer (ExTL; WMO, 2003). These processes alter the composition of the
polar LMS. This is relevant to climate in various ways, for example: The radiation budget is changed through the introduction of water vapour (Forster and Shine, 2002) or methane (e.g. Riese et al., 2012); these species, however, also alter the chlorine budget in gas phase or heterogeneous reactions, with implicit effects on ozone. Furthermore, ozone may be diluted – beyond removal by chemical processing – through horizontal mixing and isentropic transport towards lower latitudes (Knudsen et al., 1998; Knudsen and Grooß, 2000).

The extratropical LMS regions are particularly sensitive to changes in ozone loading (Riese et al., 2012). As a consequence, it is necessary to resolve the physical and chemical processes in and around the LMS in detail in order to assess the implications on climate change. Climate model intercomparison (Bekki et al., 2013) and sensitivity studies (e.g. Xia et al., 2017) find the most severe disagreements in projected ozone change and ozone-climate impact within the polar LMS. In the northern hemisphere, even the sign of change is undetermined. This is the consequence of unresolved transport of ozone and counteracting
effects from catalytic depletion and stratospheric cooling during climate change, where some transport and chemical processes



may have not been implemented into the models in all detail. In particular, Khosrawi et al. (2017) have shown recently that the climate model ECHAM5/MESSy Atmospheric Chemistry (EMAC) underestimates stratospheric downward transport in the polar vortex and potentially inherits uncertainties in chlorine chemistry kinetics. The lack of observations inside the polar LMS may contribute to the enhanced uncertainty. Access is challenging, as satellite data products, e.g. from the Microwave Limb

Sounder (MLS) or the Atmospheric Chemistry Instrument-Fourier Transform Spectrometer (ACE-FTS), do not achieve the high vertical resolution necessary to resolve strong chemical gradients near the tropopause (e.g. Livesey et al., 2017; Mahieu et al., 2008; Wolff et al., 2008). On the other hand, there have been numerous airborne activities in the Arctic LMS, but only few airborne (Bonne et al., 2000; Wilmouth et al., 2006) or balloon-borne (e.g. Wetzel et al., 2015) measurement platforms that have yet sampled the polar LMS chlorine budget. Several studies focused on ozone depleting ClO and the dimer ClOOCl

(Vogel et al., 2003; von Hobe et al., 2005; Sumińska-Ebersoldt et al., 2012) and heterogeneous chlorine chemistry (Wegner et al., 2012; Wohltmann et al., 2013; summary in von Hobe et al., 2013). A comprehensive in situ sampling of chlorine species in the Arctic lower stratosphere was conducted in a dedicated mission in winter 1999/2000 using the ER-2 high altitude aircraft (Wilmouth et al., 2006). Generally, measurements from aircraft are able to combine high spatial resolution, a benefit in the manifold environment near the tropopause, with sufficient temporal coverage in a suitable deployment.

The work at hand is aimed at presenting a new high resolution and high accuracy in situ dataset of Arctic LMS chlorine chemistry that complements satellite products at lower altitudes and may be a useful reference on which to test model simulations. Section 2 introduces the aircraft mission, instrumentation, model and vortex identification methods. Chlorine measurements are intercompared as a consistency check in unperturbed stratospheric conditions. The temporal evolution of chlorine partitioning throughout the winter is presented and discussed in Sec. 3. Section 4 concludes the study.

## 2   Measurements, model and methods

### 2.1   Activities within the POLSTRACC mission

The polar LMS during the Arctic winter 2015/2016 has been probed within the POLar STRAtosphere in a Changing Climate (POLSTRACC) aircraft campaign, using the German High Altitude and LOng range research aircraft (HALO). The mission objectives comprise studying (i) the structure, composition and dynamics of the Arctic winter LMS, (ii) chemical processes

that affect ozone in the Arctic winter upper troposphere/lower stratosphere (UTLS), (iii) PSCs and de-/nitrification and (iv) cirrus clouds in the Arctic UTLS. In a joint effort for the missions POLSTRACC/Gravity Wave Life Cycle Experiment (GW-LCYCLE II)/Seasonality of Air mass transport and origin in the Lowermost Stratosphere (SALSA) (combined under the acronym "PGS"), HALO performed a total of 18 science flights. The long range capability has been extensively used during the 156 flight hours. Spreading from the campaign bases in Oberpfaffenhofen (EDMO), Germany and Kiruna (ESNQ), Sweden,

the flight tracks span a region between 25° N – 87° N and 80° W – 28° E over Europe, the North Atlantic and Greenland (map in Fig. 1). Potential temperatures at flight level reached a maximum of 411 K. Chronologically, the observations cover most of the evolution of the polar vortex: two flights sampled the early winter formation of the vortex in mid December; the first major phase in January (eight flights) was dedicated to the established vortex, while the second major phase from end of February to





mid March (eight flights) witnessed the late stages of vortex evolution including the major final warming (MFW) and vortex split. HALO was equipped with a set of in situ and remote sensing instruments, measuring atmospheric quantities at, below, above and alongside the aircraft position.

The Arctic polar vortex during the winter 2015/2016 has already been subject to a number of studies, most of them directly
related to POLSTRACC activities: Overviews of the evolution of the polar vortex are given by Manney and Lawrence (2016), Matthias et al. (2016) and Khosrawi et al. (2017) including satellite, reanalysis and model perspectives. Concurrently, the authors point out the exceptional strength of the vortex due to reduced planetary wave activity, accompanied by record low temperatures and extensive denitrification, dehydration and chlorine activation. Record ozone loss comparable to the extreme winter 2010/2011 (e.g. Sinnhuber et al., 2011) was only prevented by an early onset of sudden stratospheric warming (SSW)
and vortex splitting events. The study by Khosrawi et al. (2017) also points to deficiencies in the representation of downward transport and chlorine chemistry in the chemistry-climate model EMAC, that may significantly impact chlorine partitioning in the LMS. The formation of PSCs has been investigated by Voigt et al. (2018) with a focus on the pathways of ice PSC nucleation and Dörnbrack et al. (2017) studied a case of gravity wave-induced PSC formation. A novel approach of three-dimensional visualisation of gravity waves from remote tomographic sampling with the Gimballed Limb Observer for Radiance Imaging
of the Atmosphere (GLORIA) has been presented by Krisch et al. (2017), while Johansson et al. (2018) examined how the same instrument consistently fills the gap between high coverage spaceborne sampling and accurate high resolution in situ measurements. Krause et al. (2018) used chemical tracers to reveal the influence of mixing at the lower vortex edge on the age spectrum of the air.

## 2.2 Airborne in situ measurements

This section provides an overview of the measurement principles of the in situ instruments used in the present work. In addition to the dedicated trace gas measurements, the BAsic HALO Measurement And Sensor system (BAHAMAS) provides meteorological parameters along the flight trajectory (Krautstrunk and Giez, 2012; Giez et al., 2017).

### 2.2.1 Chlorine measurements with AIMS

The Airborne chemical Ionization Mass Spectrometer AIMS features a pressure-controlled electrical discharge source and in-
flight calibration capability (Jurkat et al., 2016). Two different operational modes can be selected, enabling either the detection of low water vapour concentrations (Kaufmann et al., 2016) or measuring the trace gases $SO_2$, HCl, $HNO_3$, HONO and $ClONO_2$ (Jurkat et al., 2016). In the latter configuration, $SF_5^-$ reagent ions are used for the ionization of the desired trace gas molecules (Marcy et al., 2005; Voigt et al., 2010; Jurkat et al., 2010). AIMS in trace gas configuration was deployed earlier onboard the HALO and Falcon research aircrafts for measurements in the troposphere and lower stratosphere from the
northern mid-latitudes down to the Antarctic continent (Voigt et al., 2014; Jurkat et al., 2014, 2017). In this work, we present measurements of HCl and $ClONO_2$ obtained during the combined PGS campaign. $HNO_3$ and $SO_2$ were also measured, but are not in the focus of this study. In-flight calibrations were performed for HCl and $HNO_3$. Calibration uses an azeotrope solution of HCl or $HNO_3$ in water, each sealed in a permeation tube and held at constant temperature. Through the resulting constant





vapour pressure of the liquid, a specified current of gaseous HCl or $HNO_3$ can leave the tube via a membrane. Together with a nitrogen carrier gas flow, it is then added to the atmospheric air sucked through the inlet during stable background conditions. This way, any interaction of trace molecules with the tubing between the inlet and the instrument is already taken into account by the calibration. $ClONO_2$ could not be calibrated directly due to the lack of an accurate source to stably generate the necessary

amounts of calibration gas. Instead, we made use of the fact that the kinetics of the fluoride transfer reactions, carrying $F^-$ from the $SF_5^-$ reagent ions to the trace gas molecules, are similar for $HNO_3$ and $ClONO_2$ due to the molecular similarity. Thus, $ClONO_2$ values could be calculated via the calibration of $HNO_3$, using literature values for the relative sensitivities of $HNO_3$ and $ClONO_2$ (Marcy et al., 2005) and taking into account the mass discrimination between the two species (Jurkat et al., 2016). AIMS measurements were performed at a 1.7 s time resolution, equivalent to roughly 350 m horizontal resolution. Smoothing

the raw data in a 17 s (10 data points) running average window (Jurkat et al., 2016, 2017) yields detection limits of 6–12 parts per trillion (pptv) of molar mixing ratio and a 10–15 % precision for both HCl and $ClONO_2$, with an accuracy of 12 % for HCl and 20 % for $ClONO_2$.

### 2.2.2 $N_2O$ and $Cl_y$ from in situ measurements

Measurements of $N_2O$ were performed by the TRIHOP instrument, a three-channel quantum cascade laser infrared absorption

spectrometer (Müller et al., 2016). The integration time is 1.5 s with a precision of 1.84 parts per billion (ppbv) (Krause et al., 2018). After linear drift correction the total uncertainty of $N_2O$ during the POLSTRACC mission is estimated to be 2.5 ppbv. Total inorganic chlorine ($Cl_y$) is inferred from a correlation to dichlorodifluoromethane (CFC-12). The (stratospheric) correlation has been established using cryogenic whole air sampling on two balloon flights inside the Arctic polar vortex in 2009 and 2011 (Wetzel et al., 2015). It was updated to consider the tropospheric conditions at the time of the campaign,

employing the method described in Plumb et al. (1999). CFC-12, in turn, was measured by the GhOST GC-MS instrument (Sala et al., 2014) with a precision of 0.2 % at an average time resolution of 1 min.

### 2.2.3 Inferred $ClO_x$

As a proxy for active (ozone depleting) chlorine derived from the above observational data, the quantity $Cl_y - (HCl + ClONO_2)$ is calculated and hereafter termed "$ClO_x$ from measurements". This quantity features an accuracy of 23 % or 20 pptv, whichever

is larger, and a precision of 14–21 %, plus a possible systematic error from the inferred $Cl_y$ that may in fact be the main source of uncertainty. Negative values of $ClO_x$ can occur if the errors of the original tracer measurements add up adversely, or if the stratospheric correlation between $Cl_y$ and CFC-12 is altered unforeseen, e.g. near the tropopause.

### 2.2.4 Instrument comparison

In order to assess the data quality, simultaneously measured quantities from the AIMS and GhOST instruments are compared

in unperturbed conditions where the sum of reservoir gases equals the amount of $Cl_y$. To this end, Fig. 2 displays a comparison of AIMS and GhOST data for Flight 04 of the PGS campaign. This was the first science flight carried out on 17 December





2015. It started in Oberpfaffenhofen (EDMO) and was headed to the Atlantic Ocean north of Ireland, up to 59.8° N, in order to horizontally cut the tropopause at high altitude in north-south direction. Polar vortex air was only sparsely hit, because the vortex at this time of the year was higher than the maximum altitude of HALO (Matthias et al., 2016). At this early stage of the vortex development, no activated chlorine is expected and as a consequence, inorganic chlorine is should be partitioned entirely

into the reservoir species HCl and ClONO$_2$. The simultaneous measurements of AIMS and GhOST are compared in Fig.2b. Gaps in the AIMS data occur during in-flight calibration and background measurements. AIMS measurements were averaged to the lower time resolution of GhOST to numerically compare the measurements of both instruments. Figure 2a shows (HCl + ClONO$_2$) - Cl$_y$ relative to Cl$_y$ as a function of time. For the most part, the data agree within $\pm\,20\,\%$, which is well within the measurement uncertainty. Larger deviations are only found in tropospheric sections of the flight, where the correlation of

CFC-12 and Cl$_y$ is less robust since it varies with tropospheric sources. As shown in Fig. 2c, the correlation between (HCl + ClONO$_2$) and Cl$_y$ for this flight is linear over the whole concentration range with $R^2 = 0.989$. Overall, the agreement is excellent, given that this comparison aggregates three independently measured and one derived trace gases. Similarly, but not shown here, ClONO$_2$ (Johansson et al., 2018) and HNO$_3$ data (e.g. Ungermann et al., 2015) from AIMS compare well to other in situ measurements and remote sensing products aboard HALO.

## 2.3    Chemical transport model

Simulations are performed with the Chemical Lagrangian Model of the Stratosphere (CLaMS) that is described elsewhere (Grooß et al., 2014, and references therein). Unlike most other Eulerian models, the Lagrangian chemical transport model CLaMS calculates the chemical composition along airparcels irregularly distributed over space that follow individually their trajectories. The underlying wind and temperature data are taken from ECMWF ERA-Interim data (Dee et al., 2011). Initiali-

sation and boundary conditions of the model simulation for the winder 2015/2016 are described by Grooß et al. (2018).

Typically, the model output is written every day at 12:00 UTC. For interpolation to the observation locations and times, a Lagrangian mapping was used, employing back trajectories from the desired positions and times to the previous day. Interpolated from the model output, the CLaMS chemistry module is integrated forward to the observaion point. With that procedure, the chemical composition (including the simulated chlorine partitioning) at the observation location and time is determined

from the model.

### 2.4    Vortex identification

#### 2.4.1    Identification of vortex air by in situ measurements

The study of chlorine partitioning within this work is intended to focus solely on air masses that can be attributed to the Arctic polar stratospheric vortex. A common means for vortex identification is the Nash criterion (Nash et al., 1996), essentially

defining the location of the vortex edge by the maximum gradient of potential vorticity (PV) on isentropic surfaces. In this work, however, the method of Greenblatt et al. (2002) is applied where vortex air is identified by a tight correlation between the inert tracer N$_2$O and potential temperature. Using this method, the subsequent analysis benefits from the high resolution





of $N_2O$ and potential temperature measurements in contrast to reanalysis of PV fields. Thus, also small-scale patterns at a size of several kilometres or filamentary structures can be assigned correctly. Figure 3 aggregates all $N_2O$ data points from the PGS campaign versus potential temperature ($\theta$). The diagram is constrained to values above 320 K, focusing thus on the stratosphere above the ExTL at high latitudes. The observed $N_2O{:}\theta$ profile narrows towards the tropopause, whereas it widens

towards higher altitudes, showcasing the wide variety of air masses sampled during the flights. The labels in the upper part of Fig. 3 roughly indicate the region of the sampled air masses. Generally three regimes can be distinguished: the vortex regime is found at the inclined left edge of the profile characterised by a strong vertical gradient of $N_2O$ mixing ratio, which connects the tropospheric source region to the photochemical sink in the middle stratosphere (Schmeltekopf et al., 1977). The gradient is maintained by the stratification and isolation of the polar vortex, while isentropic homogeneity leads to a compact form of

the profile. At the other extreme, air from mid latitudes exhibits a more variable gradient in $N_2O$ due to altering influence from tropical or polar air. The lower $N_2O$ mixing ratios inside the vortex, on constant isentropes, result from the large-scale descent of $N_2O$-poor air from higher altitudes. Both regimes merge towards the troposphere, where the vortex transport barrier vanishes. In between at higher altitudes, we find the boundary region with a continuous shift between vortex and mid latitude contributions.

A flight on 26 February 2016 (flight 14) provides a reference profile, reaching well into the vortex and capturing dynamically unperturbed vortex conditions. The flight exhibits a very compact relationship between $N_2O$ and $\theta$ for a wide range of potential temperatures at the lower left edge of the data set in Fig. 3 (blue points). $N_2O$ measurements from this flight are binned in 5 K intervals of $\theta$, and the mean values are taken as the vortex reference profile (triangles in Fig. 3). Observations include also outside-vortex air as indicated by the light blue points between 345 K and 360 K, so the vortex reference points in this range

are determined by a quadratic interpolation between the adjoining sections. Beyond 395 K, the vortex reference profile is set by linear extrapolation. A symmetric envelope is placed around the vortex reference profile. The width is determined for each flight individually by the maximum deviation of $N_2O$ data points below the vortex reference value in each bin. All data points lying within the envelope are then considered to belong to vortex air masses (dark points in Fig. 3). This procedure is chosen to sensibly include a high number of measurements in the vortex regime, accounting for instrumental noise and small atmospheric

variability while the criterion is not weakened too much. Hereafter, the term "vortex air" is meant to refer to measurements that fulfil the vortex criterion above 320 K. Equally, the term "extra-vortex air" is the complimentary set of measurements above 320 K.

     Although the determined vortex reference correlation seems to be appropriate for the whole data set at first glance, we have to account for diabatic descent during the three month campaign phase. Satellite data of $N_2O$ (Manney and Lawrence, 2016)

and analysis of the in situ $N_2O$ data (Krause et al., 2018) indicate that below $\theta = 450$ K, diabatic descent, i.e. descent of air masses versus isentropic surfaces, was strongest before the end of December 2015. Accordingly, the vortex reference profile from Flight 14 was lifted by 15 K only for the two December flights, while left unchanged for the rest of the flights.



### 2.4.2 Identification of vortex air in the model

For model data interpolated along the flight tracks, the measured $N_2O{:}\theta$ vortex criterion from Sec. 2.4.1 is applied unchanged, in order to achieve an optimal comparison to the measurement data. For vortex-wide averages, all model data with equivalent latitude $\Phi_e$ greater than $65°$ N are included. This choice results in a constant area (in square kilometres) and does not reflect the variability of the Nash criterion (Grooß and Müller, 2007), but – from experience – matches it quite well for a fully developed polar vortex, whereas the vortex area is overestimated during very early and late stages.

## 3 Results: Chlorine partitioning throughout the Arctic winter 2015/2016

### 3.1 Overview of vortex air sampling

The flights of the PGS campaign were conducted from 17 December 2015 until 18 March 2016, covering several stages of evolution of the Arctic polar vortex. The map in Fig. 1 depicts vortex air sampling sections of the conducted flights in blue, as opposed to extra-vortex air sections in black. Two different patterns can be clearly distinguished: On the one hand, there are flights with long sections entirely inside the vortex, resulting in a good sampling statistics and providing insight to intra-vortex variability. On the other hand, many flights exhibit only a patchy sampling of vortex air whenever other objectives of the PGS campaign were pursued. As an example, the PV maps in Figure 4 illustrate that a compact vortex signature is visible at 50 hPa (above 20 km altitude), whereas it becomes more spread-out and filamented below. On that particular day, vortex air measurements could be conducted over Greenland and northern Canada near the PV maximum on 150 hPa, which is the approximate flight altitude. Split-off parts of vortex air could also be found at latitudes down to $42.7°$ N towards the end of the campaign in early spring.

Figure 5 gives an overview of the temporal sampling of vortex air above different potential temperatures $\theta$ over the whole PGS campaign. Therefore, the number of individual (HCl) measurements during each flight is summed up and depicted in columns. The colour code indicates the vortex air encounters divided in measurements above a certain potential temperature. Extensive vortex air sampling was performed primarily at the end of the first main phase and during the second main phase, in concert with the gradual descent of higher vortex air masses to flight altitude. Specifically, vortex air encounters represent between 3 % and 95 % of flight time of individual flights. In situ sampling was focused on $\theta > 340$ K for most flights, keeping significant distance to the tropopause. Vortex air above $\theta = 380$ K was only sampled as of 26 February 2016 when mainly adiabatic transport brought air masses with such high potential temperatures within reach of the HALO aircraft. The 400 K isentrope was only crossed during three occasions in March with a total time of about 10 min inside vortex air.

### 3.2 Measured evolution of chlorine gases during the winter

The evolution of inorganic chlorine partitioning in the lowermost Arctic polar vortex over a period of three months is assessed by means of daily statistics, which is performed by calculating averages, standard deviation and quantiles from all "vortex air" data points during individual research flights, in order to get a reasonable statistical sample size. This allows the study





of changes in trace gas concentrations on a time scale of days. Beforehand, the data are binned into four layers of potential temperature, each spanning 20 K, to introduce a coarse quasi-vertical coordinate. The panels a–d in Fig. 6 display the mean of the measured distributions for the individual HALO flights. Colours indicate the $\theta$ layers. The layer 320–340 K is omitted for clarity as it shows almost no difference to the 340–360 K layer. The panels e–g are similar to panels a–c, but display the

relative abundance with respect to $Cl_y$ instead of absolute mixing ratios.

In December 2015, the measurements indicate that inorganic chlorine is partitioned almost entirely into the reservoir species HCl and $ClONO_2$ below the 380 K isentropic surface. With almost 80 % contribution to $Cl_y$, the photochemically stable HCl is predominant, whereas $ClONO_2$ ranges with less than 0.12 ppbv below or near the detection limit. This ratio is common in unperturbed conditions (e.g. Santee et al., 2008). At this early stage of the polar vortex, no $ClO_x$ has been detected.

The partitioning changes in January 2016, where the vortex is fully developed and temperatures are low enough to enable heterogeneous chemistry. The chlorine activation manifests itself in several ways in the observations: The absolute HCl mixing ratio does not show a general trend through early February below 380 K, but in terms of relative partitioning, HCl is clearly depleted with respect to December conditions. In addition, HCl exhibits intermediate minima at the length of days with a decrease by up to 0.21 ppbv, as well as a final minimum on 2 February, that may indicate episodes of enhanced chlorine

activation. The lowest HCl mixing ratio measured above 360 K is 0.17 ppbv on 20 January. Correspondingly, at least two phases of enhanced active chlorine can be identified, in mid and late January/early February. On 18 January, measurements suggest that up to 75 % or 0.58 ppbv of $Cl_y$ are activated below 380 K. These observations correspond with periods of high occurrence of ice and NAT PSCs (Voigt et al., 2018) which provide surfaces for heterogeneous chlorine activation. Enhanced mixing ratios of $ClONO_2$ up to 0.52 ppbv show how chlorine is repartitioned once sufficient amounts of ClO have been

produced. The apparent rise in mean total $Cl_y$ mixing ratio between December and January at 360–380 K from 0.44 to about 0.85 ppbv reflects the diabatic cooling that brings higher-level air masses to lower potential temperatures.

Large parts of the lower stratosphere could be sampled during the second main phase of the HALO deployment in late February and March, which covers the late stages of vortex evolution including the major final warming (MFW). The measurements show significant chlorine activation at the beginning of the second campaign phase when HALO could sample for the

first time air masses above 380 K potential temperature, where a peak amount of 1.15 ppbv $ClO_x$ at the flight altitude of 13 km was found. The MFW on 5 March (Manney and Lawrence, 2016) observably terminated heterogeneous chlorine activation reactions, with remaining $ClO_x$ amounts no greater than 0.20 ppbv. The recovery of the reservoir species varies strongly across the specified isentropic layers (supported by dashed lines in 6e–f): below 360 K, a gradual increase in HCl up to 0.58 ppbv throughout March can be seen, whereas $ClONO_2$ soon returns to concentrations below the detection limit. The stratospheric

warming makes higher $\theta$-level accessible to HALO, and above 380 K, recovery pathways are different: depleted HCl starts to recover, but this positive trend is halted and eventually reversed by early to mid March. Meanwhile $ClONO_2$ increases strongly above the level of HCl mixing ratios. Between 360–380 K (red data), observations are not very clear about the overall evolution, especially of $ClONO_2$. Also, negative values for $ClO_x$ are calculated. While these are still within the uncertainty limits, this isentropic layer seems to contain a strong vertical variability of chemical constituents involved in chlorine recovery,

communicating the different behaviour below and above. The vertical sampling of HALO may introduce a bias, which is also



obvious in the enhanced variability of mean $Cl_y$ values in the layers above 360 K. This is investigated further by the model intercomparison below.

Figure 7 illustrates the concept of differential deactivation pathways in this late winter period in a different manner. There, the partitioning of $Cl_y$ is shown in a ternary diagram of HCl, $ClONO_2$ and $ClO_x$ fractions. The dots mark individual measurements, the same that the daily averages in Fig. 6e–g were based on. The solid arrows run along the temporal evolution of the daily averaged measurements and show the two pathways below 360 K (black) and above 380 K (blue), diverging towards HCl or $ClONO_2$, as well as the 360–380 K isentropic layer (red) sandwiched in between. Using the supporting green isolines of the HCl/$ClONO_2$ fraction, we observe that below 360 K, the partitioning between the reservoir species is more HCl-heavy and after recovery, HCl is three times more abundant than $ClONO_2$. Above 380 K, in contrast, the reservoir species evolve from a nearly 1:1 partitioning to a state with twice as much $ClONO_2$ relative to HCl.

## 3.3 Comparison of measured and CLaMS-modelled data

Simulations by the Lagrangian CTM CLaMS were performed to put the observations into a broader context, and to investigate sampling biases potentially caused by the coverage of the aircraft observations. To assess the accuracy of the model, first the model results were interpolated to the location and time of the observations employing the trajectory mapping described above. Figure 8 displays the model results (crosses) alongside the observations sampled with the same vortex criteria and averaging procedures. As for HCl, the model generally follows the trends of the measurement data. Before February, however, there is a clear high bias by 0.13–0.20 ppbv in almost all model HCl data which cannot be systematically seen after the break. Having ruled out technical changes on the simulation and on the instrumental sides during this time, it seems as if we can observe a model bias that is known from satellite comparisons at higher altitudes in the dark polar vortices, a problem that is observed by other models as well (Wohltmann et al., 2017; Grooß et al., 2018). Our intercomparison of model and in situ data supports the rationale that chemistry-climate models struggle in reproducing the observed chlorine partitioning in the dark winter months, where some unknown process for HCl removal is lacking. Here we extend the previous observations by Grooß et al. (2018) to lower altitudes. Consistently, the discrepancy is absent as soon as sufficient sunlight returns towards the end of the winter. An overestimation of HCl is partly reflected in an overestimation of $Cl_y$ and underestimated $ClONO_2$ and $ClO_x$ concentrations.

$ClONO_2$ is better represented in the model below $\theta = 360$ K, whereas there are indications of a slight underestimation on higher isentropes. This may be induced in part by the HCl high bias until midwinter, or by generally underestimated $Cl_y$ mixing ratios recognizable above 380 K. The model is able to produce the observed change in recovery across the different isentropes. The modelled vertical shift of HCl versus $ClONO_2$ recovery in March between 360–400 K is obviously subject to a sampling bias just like the measurements, but at the same time does not always follow the observational data. This deviation is probably caused by a lack of vertical resolution in the meteorological data fields that prevents the model from estimating the high vertical gradients in the atmosphere in all detail.

Beyond this direct intercomparison, model data can help explore to what extent the observations along the aircraft's flight track are representative of the entire polar vortex. Therefore, the lines in Fig. 9 display the modelled concentration of the chlorine species in a vortex-wide average at the different isentropic levels. In general the local aircraft observations reflect



the mean chlorine partitioning in the vortex LMS quite well. The phases when activated chlorine can be found are consistent, whilst enhanced local variation of $ClO_x$ in January is not projected into the vortex averages. Vortex-averaged HCl mixing ratios clearly suffer from the deviation during the dark episode. Toward the end of the winter, the vortex averages include air masses from lower latitudes at increased frequency. This is accompanied, for example, by a sudden drop in modelled mean ozone (not

shown), which would explain the lower values of all chlorine species, compared to the measurements in the local remainders of vortex air.

### 3.4   Variability of chlorine partitioning

To assess the validity of using daily averaged mixing ratios as in the previous sections, Fig. 10 displays in addition to mean values the extrema and quartiles from the statistics studied hitherto, where the 360–380 K layer is taken as an example that

stretches over all measurement phases. Intermediate data spread, as seen in most January flights, reflects the small and steady variation through the depth of the layer, while trends between consecutive flights are clearly visible. There, the statistics are based on a solid sample size, and these are the periods where measurements and model data match best. A very low variability, such as for the first December flight or for two flights on 10 and 13 March stems from very few vortex sampling points in this layer, rendering these values less representative. Consistently, the model data interpolated along the flight track deviates

strongest from the measurements on these dates, as visible in the HCl and $ClONO_2$ results. Very high variability, as seen on 26 February and 6 March hints at enhanced differential processing or transport within the sampled air masses.

    The standard deviation of the vortex-averaged statistics (grey shading in Fig. 10a–d) reflects the intra-vortex atmospheric variability. Without accouting for potential systematic biases, two thirds or 67 % of the observational data are found within this spread, this demonstrates the joint performance of the measurements and the model at reproducing vortex air characteristics on

single aircraft flight tracks.

### 3.5   Discussion

In this section, our observed evolution of chlorine partitioning during the winter 2015/2016 is compared to previous studies using in situ and remote sensing data.

    The Arctic polar vortex during winter 2015/2016 has been studied by Manney and Lawrence (2016), using measurements

from the Aura MLS instrument. As for chlorine species, they retrieved HCl and ClO at a vertical resolution of 2.5–6 km down to $\theta = 390$ K. Therefore, the satellite data only have a small overlap with our in situ observations in March. Nevertheless, the depletion and recovery of HCl as well as the occurrences of active chlorine, as reported in section 3.2, consistently extend the remote sensing observations to lower altitudes, and show that chlorine activation is generally not limited to the higher altitudes inside the Arctic polar vortex.

Based on measurements on the ER-2 aircraft within the SOLVE/THESEO mission, Wilmouth et al. (2006) drew a very comprehensive picture of inorganic chlorine partitioning in the Arctic polar vortex of 1999/2000, where low temperatures (Manney and Sabutis, 2000), high PSC presence and large chemical ozone loss (Rex et al., 2002) were observed. At isentropes around 440 K, $ClO_x/Cl_y$ reached up to 90 % in January and HCl accounted for the remaining 10 %, whereas $ClONO_2$ was




not detectable. Thus, the higher degree of activation at higher potential temperatures, as compared to the PGS measurements, seems to come mainly at the expense of $ClONO_2$. From their late winter budget analysis, Wilmouth et al. (2006) suggest that HCl recovers at a rate similar to or higher than the $ClONO_2$ recovery rate. They state that the evolution of inorganic chlorine partitioning in the Arctic is rather variable. Whether the chlorine deactivation into $ClONO_2$ or HCl is favoured,

depends critically on the mixing ratios of ozone and $NO_x$ (Grooß et al., 2005). Our late winter measurements are in line with observations during a balloon sounding of the MIPAS-B (Michelson Interferometer for Passive Atmospheric Sounding) and TELIS (TErahertz and submillimeter LImb Sounder) in the late Arctic winter 2011 (Wetzel et al., 2015). There, HCl is the dominant reservoir species below 14 km altitude, while the ratio is shifted towards $ClONO_2$ between 14 and 24 km. In an overview of stratospheric polar winter chlorine partitioning by Santee et al. (2008), based on MLS and ACE-FTS satellite data,

observations at $\theta = 500$ K in different Arctic winters show different patterns of chlorine recovery: In 2004/2005, chlorine is mainly deactivated by $ClONO_2$ formation, which rises to about 60 % at the end of the winter, similar to our observations above 380 K. Only later is chlorine slowly repartitioned into HCl through $ClONO_2$ photolysis towards the unperturbed conditions. The Arctic winter 2005/2006, instead, reveals a much faster recovery of HCl, similar to what Wilmouth et al. (2006) report and what is more reminiscent of the Antarctic stratosphere.

The general year-to-year variability in the northern hemispheric polar vortex is caused by high wave activity, which impacts the stability of the vortex and achievable cooling. On the process scale, the tradeoff between HCl or $ClONO_2$ recovery is controlled by the availability of reaction partners. $NO_2$ is needed for $ClONO_2$ formation, and may be introduced through mixing at the vortex edge (e.g. Krause et al., 2018), or through photolysis of gaseous $HNO_3$. HCl recovery is favoured if low ozone leaves enough Cl radicals to react with methane, which is typically the case in the lowermost stratosphere.

The same instrumental configuration was used earlier in a probing of Antarctic polar vortex air on 13 September 2012 at isentropes between 320 K and 385 K (Jurkat et al., 2017), where up to 40 % active chlorine were measured. Both the Arctic and Antarctic aircraft observations were made at (static) temperatures above 199 K, often even above 210 or 220 K. This is too warm for PSCs and for heterogeneous processes. Therefore, observable $ClO_x$ must have been activated beforehand, potentially with subsequent downward transport. This shows that, on both hemispheres, active chlorine can remain present also close to

regions with temperatures below the PSC threshold.

## 4   Conclusions and Outlook

The present study uses high accuracy and high resolution in situ aircraft measurements in the lowermost parts and outflow regions of the Arctic polar vortex during winter 2015/2016 to investigate the evolution of inorganic chlorine species. Various different scientific targets during the campaign resulted in a comprehensive sampling of stratospheric active chlorine species

over the course of the winter.

    The observations below $\theta = 400$ K never showe full chlorine activation. The appearance of active chlorine beyond its formation region is important as it increases the oxidation capability of the polar and high latitude UTLS and thus may impact the sensitive radiation budget through enhanced removal of ozone and methane. We highlight the difference in partitioning of




the reservoir species from the lower isentropes to the higher ones during the recovery phase, where the transition from HCl recovery to mainly $ClONO_2$ recovery is located within the band 360–380 K. The different recovery pathways are supposed to be caused by gradients in available ozone and $NO_2$.

Comparing the measurements to vortex-averaged model data, our local observations reflect the larger scale patterns of active
and repartitioned chlorine. The occurrence of active chlorine is simultaneous to activation maxima at higher altitudes, as seen in satellite data. While HALO with a ceiling altitude of 15 km was not anticipated to be an ideal aircraft for measurements of chlorine activation in polar vortex air, we here show the capabilities and potential of random sampling of the polar LMS. The CLaMS chemistry model, in conjunction with the underlying meteorological fields, performs very well in reproducing the in situ measurements, indicating that relevant homogeneous, heterogeneous and transport processes are realistically simulated.
One exception is the overestimation of HCl in the dark December and January vortex, which remains to be resolved (Grooß et al., 2018). This study helps to reveal the finer scale variations and gradients in chlorine processing in the outflow regions of the polar vortex. It may prove useful in further checks of climate models and detailed investigations on the impact of vortex-processed air on surrounding areas, also in the light of comparable earlier and future deployments of this instrumental configuration in the polar UTLS.

*Data availability.* The observational data of all HALO flights during the combined PGS campaign are available at the HALO database https://halo-db.pa.op.dlr.de. ERA-Interim reanalysis data are available at ECMWF https://www.ecmwf.int. CLaMS model data are available from the authors upon request.

*Author contributions.* AM, TJ, CV and JUG performed the study and AM wrote the manuscript. Measurements were performed by AM, TJ, SK, RH and CV (AIMS), AE (GhOST) and PH and JK (TRIHOP). JUG performed and studied the CLaMS simulations. All authors
contributed to the final manuscript.

*Competing interests.* The authors declare that they have no conflict of interest.

*Acknowledgements.* The work of AM, TJ and CV for this study was funded within the DFG HALO-SPP 1294 under contracts JU3059/1-1 and VO1504/4-1.



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





**Figure 1.** Map of flight tracks of the PGS science flights. Red stars mark the two bases of operation Oberpfaffenhofen (EDMO) at 48° N, 11° E and Kiruna (ESNQ) at 68° N, 20° E. Flight sections where polar vortex air was encountered according to the criterion from Sec. 2.4.1 are indicated in blue.





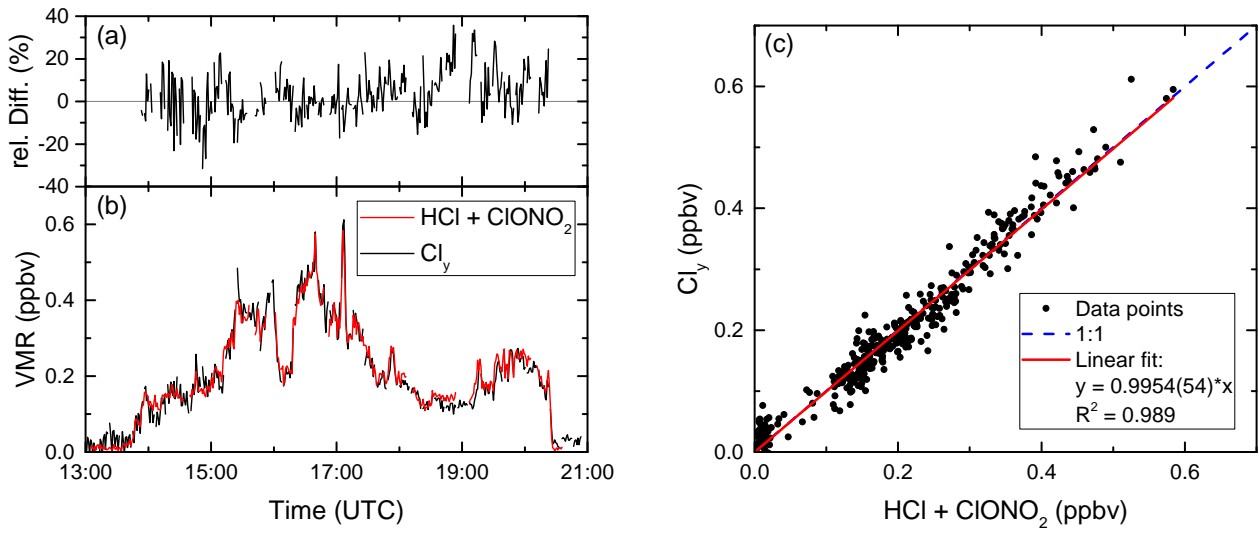

**Figure 2.** Comparison of the sum of HCl and $ClONO_2$, measured by AIMS, to $Cl_y$, inferred from GhOST measurements, during Flight 04 on 17 December 2015. (a) Difference between both curves relative to $Cl_y$ and (b) timeline of mixing ratios over the whole course of the flight. (c) Correlation between added AIMS measurements and $Cl_y$.





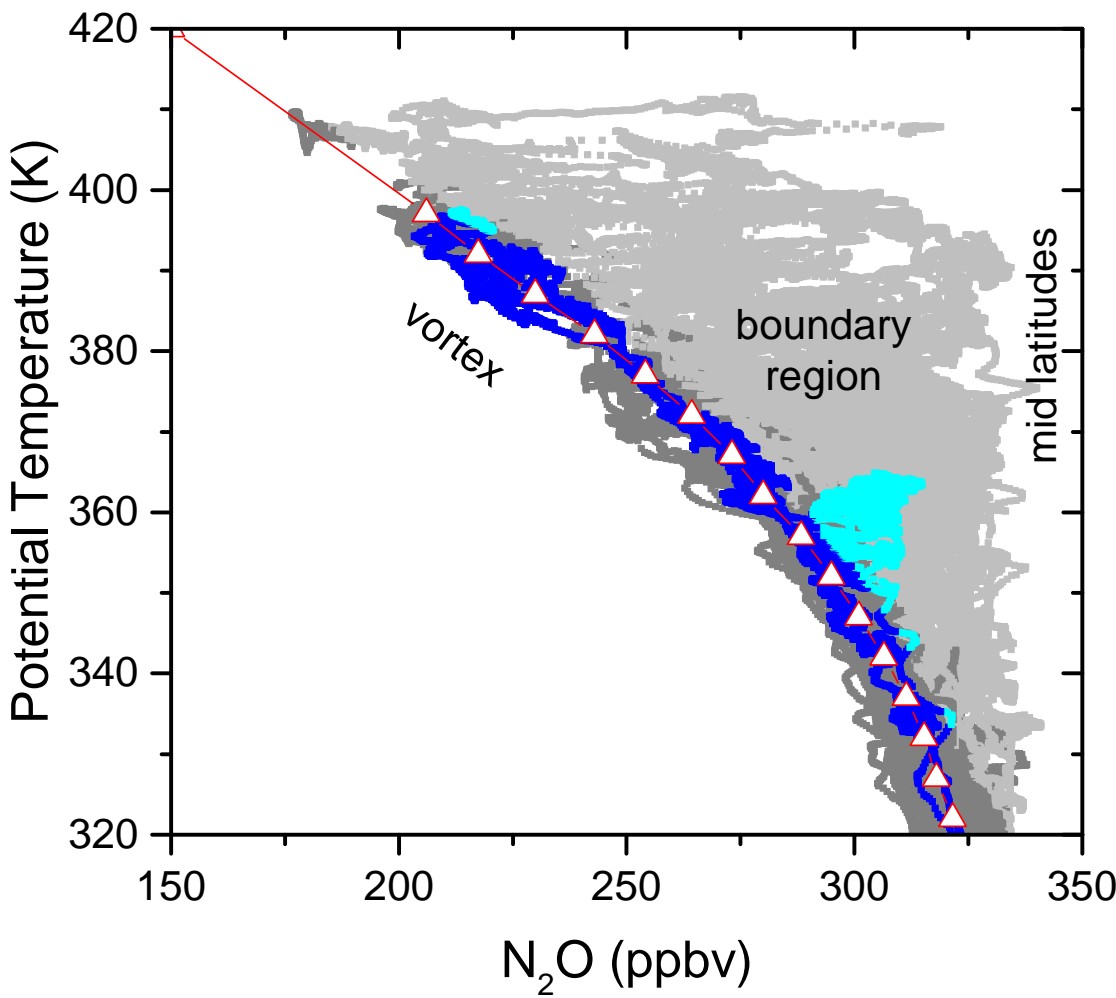

**Figure 3.** Profiles of N$_2$O versus potential temperature for the whole PGS campaign (grey) and for the selected Flight 14, of 26 February 2016 (blue), above 320 K. The derived vortex reference profile is illustrated by the triangles, connected by straight red lines. Dark and light subsets of the grey and blue points indicate the assigned vortex air and extra-vortex air property, respectively.



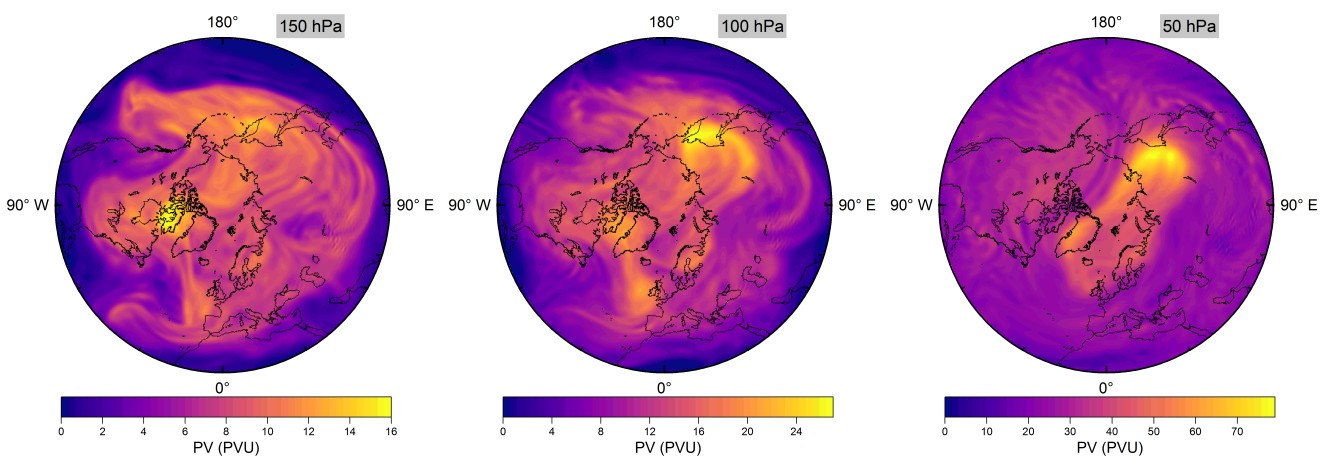

**Figure 4.** Northern hemisphere potential vorticity (PV) maps from ECMWF reanalysis at 150, 100 and 50 hPa on 26 February 2016 at 12:00 UTC. Note the different colour scales in each panel.





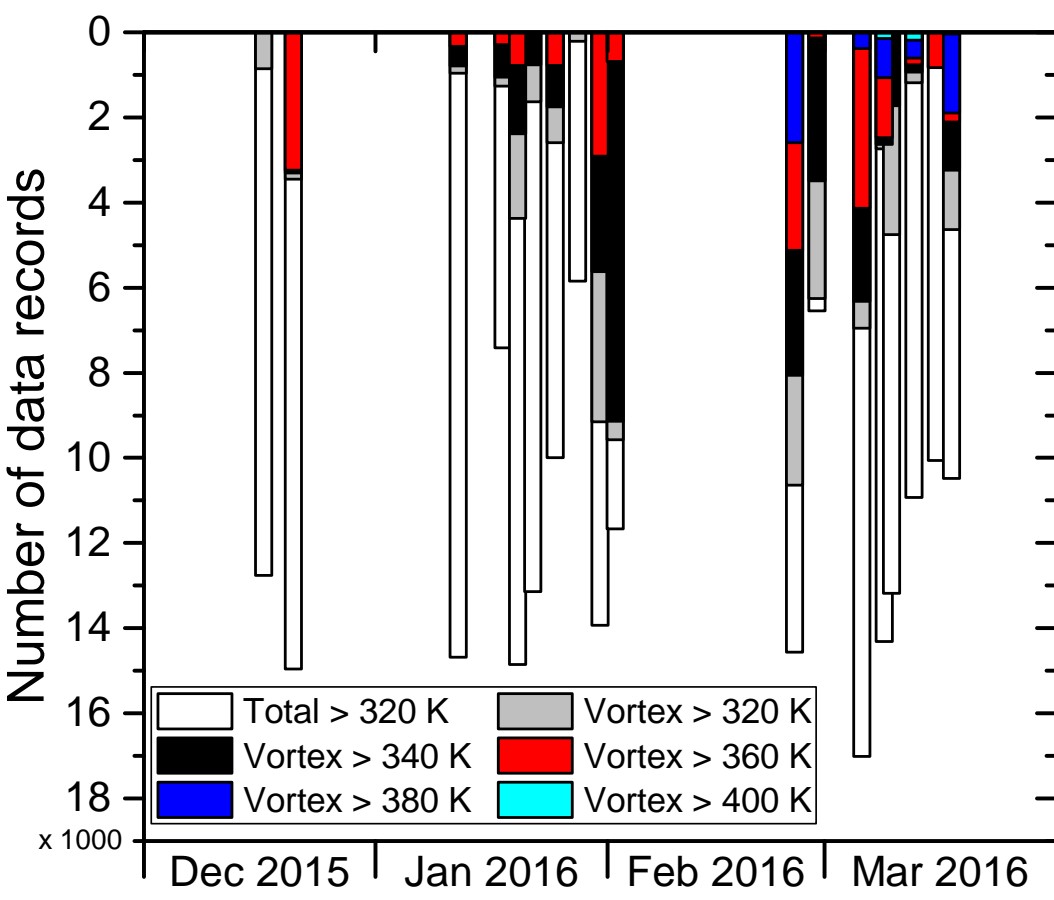

**Figure 5.** Vortex air sampling statistics of the PGS campaign, counted by the number of individual HCl measurements. Each column represents a single science flight. Coloured bars indicate portions above selected isentropes.





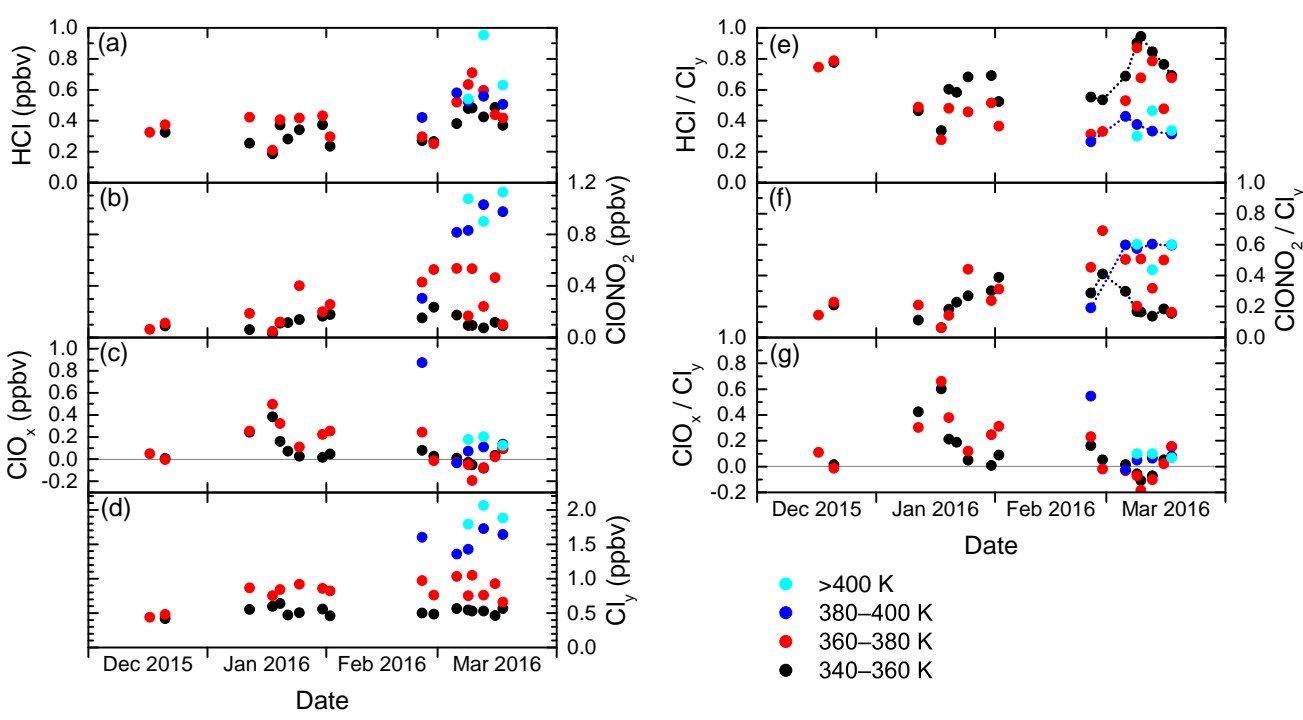

**Figure 6.** Daily averages of measured mixing ratios of HCl, ClONO$_2$, ClO$_x$ and Cl$_y$ throughout the winter 2015/2016, from data that has been labeled "intra-vortex". Left (a–d): Absolute mixing ratios. Colours are the same as in Fig. 5. Right (e–g): Similar to the left panels, but for relative contributions of HCl, ClONO$_2$ and ClO$_x$ to Cl$_y$. Some of the black and blue points are connected by dashed lines to guide the eye along the different recovery pathways.





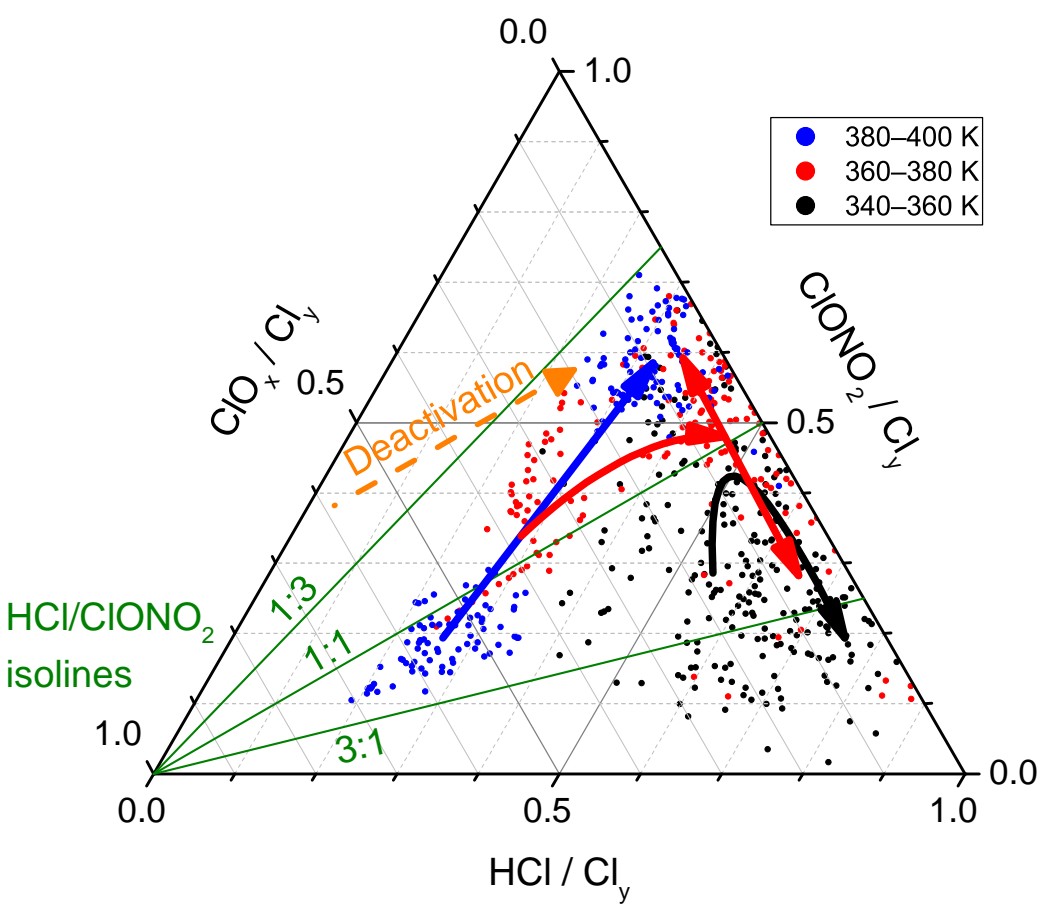

**Figure 7.** Ternary diagram of the partitioning of $Cl_y$ into HCl, $ClONO_2$ and $ClO_x$ during the second main campaign phase, from 26 February to 18 March 2016. The HCl isolines run from bottom left to top right, the $ClONO_2$ isolines horizontally, and the $ClO_x$ isolines from top left to bottom right. Dots mark individual measurements, where colours indicate the isentropic layers. Solid arrows illustrate the temporal evolution of the daily averages. As an aid to the reader, three green lines denote characteristic values of the HCl/$ClONO_2$ ratio, and the orange dashed arrow points towards decreasing values of the $ClO_x$ fraction.



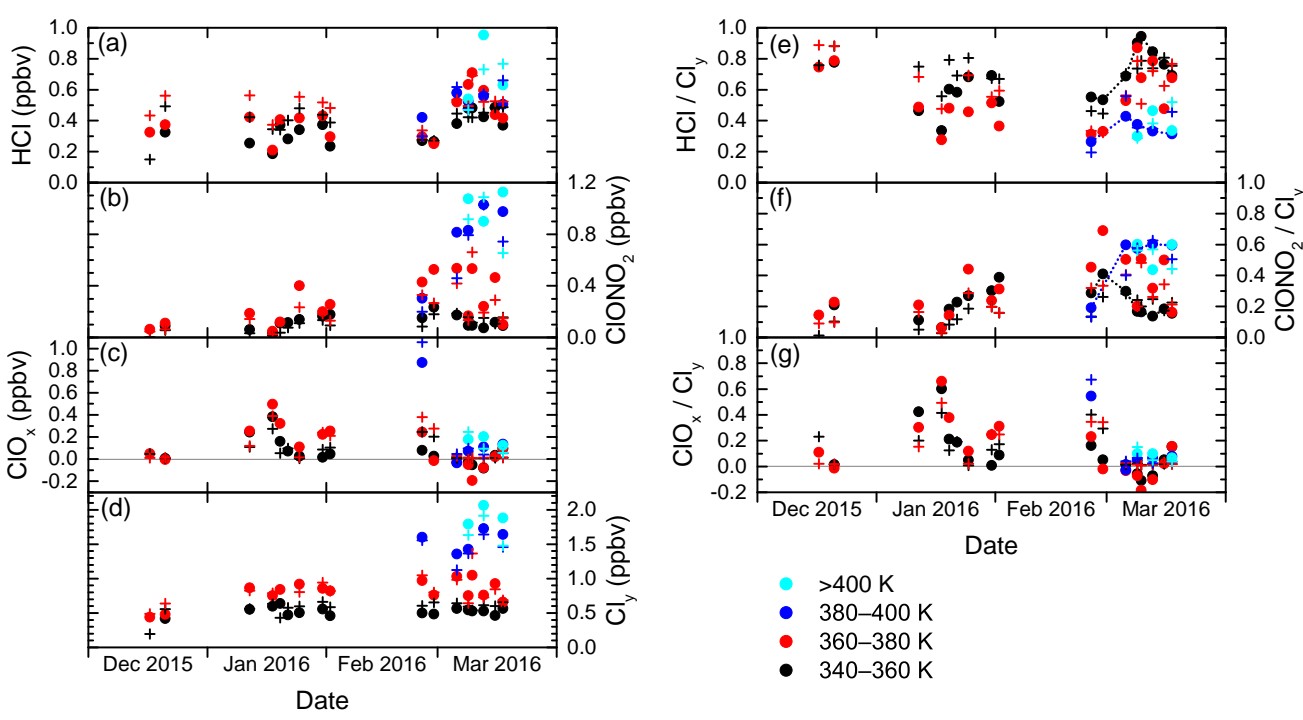

**Figure 8.** Similar to Fig. 6. In addition to averages from the measurements depicted as points, crosses denote averages from CLaMS model data interpolated at the flight track. The grey areas highlight the period where the model overestimates HCl.





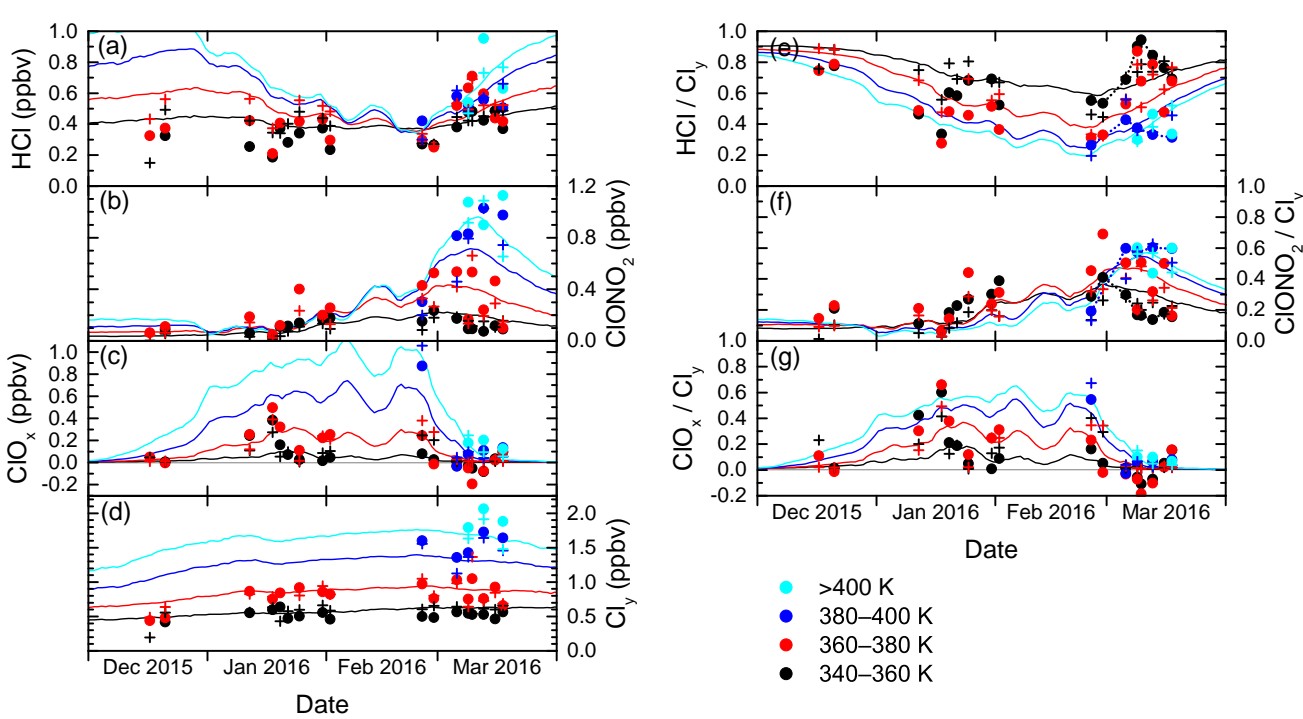

**Figure 9.** Similar to Figs. 6 and 8. In addition, lines represent whole vortex model averages.





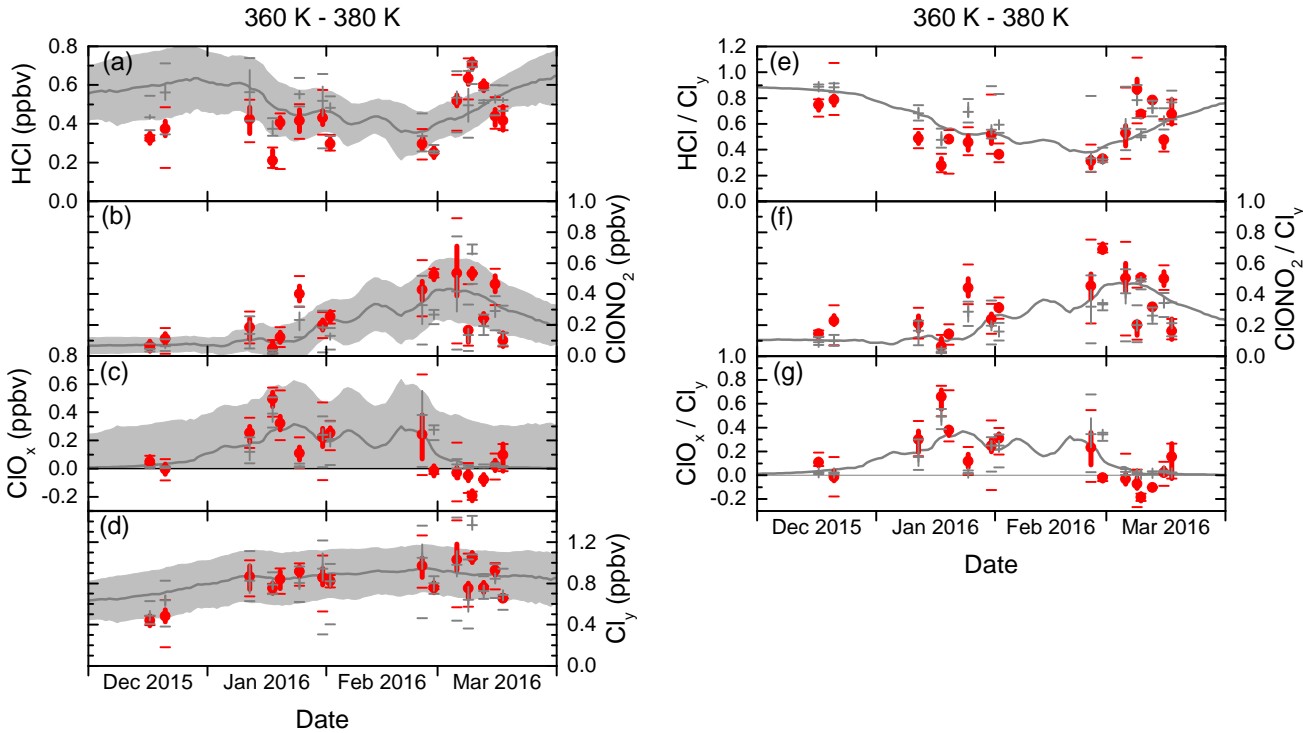

**Figure 10.** Similar to Fig. 9, with more detailed statistics of only the 360–380 K layer. Measurements are red, model data are grey. Beyond the mean values (points and crosses), vertical bars span the second and third quartiles, while horizontal bars indicate minimum and maximum values. The grey area encompasses the standard deviation from the vortex averaging (only for panels a–d).