# Peer review of "Chlorine partitioning in the lowermost Arctic vortex during the cold winter 2015/2016"

_Atmospheric Chemistry and Physics, 2019_

## Referee Comment (RC1) · Anonymous Referee #1 · 9 Jul 2019

The paper by Marsing et al. on chlorine partitioning is clearly within the scope of ACP. The paper is well structured and written, and I do not have much to criticize with respect to its scientific content. Existing work in this research field is adequately referenced. I suggest publication in ACP after consideration of the few minor comments listed below.

p3 l6: Is the term "chemical gradient" established technical language or slang? I would prefer something like "concentration gradients"

p5 l10-12: I suggest to expand somewhat on error estimation. How are the uncertainties estimated? From a statistical analysis of the data or by propagating known ingoing uncertainties through the system? If errors come from a statistical analysis, how are the systematic components estimated? Which error sources have been considered? etc.

[Figure]

p5 l11: The term "accuracy" is notoriously ambiguous. Often it is understood as an estimated of the combined random and systematic error, but equally often it is understood as the systematic part alone. One can even find official documents (GUM, ISO and others) which use contradicting definitions of accuracy. I suggest to avoid this term and to use "estimated total uncertainty" or "estimated systematic uncertainty" instead.

p6 l21: Using model output from 12:00 UTC only and interpolate data to other locations and times could cause problems in the case of a diurnal cycle. At the altitudes of interest, the diurnal cycle of ClONO2 is not very large but since this is a key condition for the applicability of this method, this assumtion should be explicitly stated.

---

## Referee Comment (RC2) · Anonymous Referee #2 · 14 Jul 2019

Review of ACP-2019-370, Marsing et al. "Chlorine partitioning in the lowermost Arctic vortex during the cold winter 2015/2016"

The manuscript by Marsing et al. describing the variation in the partitioning of chlorine in the Arctic polar vortex over the course of the winter 2015/2016 is highly topical for Atmospheric Chemistry and Physics. Overall the paper is clear and well-written. The arguments the authors make in the paper are reasonably well supported by the data they present and that data presentation is generally clear and understandable, in particular I appreciated Figure 7. I recommend the manuscript be accepted for publication, but would request the authors consider and possibly address the points below.

In section 2.2.2, it would be good for the authors to include the correlation value they have used to infer Cly from CFC-12, i.e. Cly = … CF-12
so that there is a record of what values/relationship was used.

Throughout the manuscript, I find the use of "recovery" to describe the repartitioning of Cl into reservoir species somewhat odd. I understand the usage, but it carries a viewpoint of what "normal" is, while indeed the whole repartitioning cycle between active and reservoir species is normal given the conditions. It seems like the discussion could be of "partitioning" or "repartitioning" into the reservoir species. A couple suggestions along these lines are included in the specific comments below.

Figure 4 does not seem particularly important to the argument and is only addressed as an example in a single paragraph. I would think that it could be omitted without effect on the paper.

Is there a reason for plotting the data in Figure 5 as stalactites (inverted y-axis) instead of the standard way with bars increasing upward?

The caption for Figure 8 indicates there should be grey areas to indicate the model HCl overestimation period, but the figure in my copy does not have any shaded areas.

Specific minor comments and suggestions for the text:

| | |
|---|---|
| P1 L1 | "process" seems somewhat generic—maybe use "destroy" or similar |
| P1 L6-7 | "an altitude dependent shift in the pathway of…" is a little unclear, perhaps something like "an altitude dependence of the pathway for chlorine deactivation with HCl dominating below the 380 K isentropic surface and ClONO2 becoming significant " |
| P1 L20: | "abbreviated with" → "abbreviated as" |
| P2 L1: | "Besides chlorine, bromine compounds are also important contributors to catalytic ozone destruction." |
| P3 L32-33 | suggest adding "campaign"—"first major campaign phase" |

| | |
|---|---|
| P5 L3 | "the instrument is already taken into account" could be "the instrument is already accounted for" |
| P5 L10-11 | the use "parts per trillion (pptv) of molar mixing ratio" is awkward. The "v" is used for "by volume" which is typically taken as equivalent to by mole. Options would be to delete "of" or replace with "by mole" or volume. |
| P6 L4 | "is should be partitioned"—need to use one or the other, "is" or "should be" or perhaps "is expected to be" |
| P6 L20 | "winder" → "winter" |
| P7 L1 | "of N2O … measurements" → "measurements of N2O and potential temperature" (although potential temp is calculated, not measured) |
| P7 L1 | delete "of" from "of PV", and delete "also" |
| P7 L3 | "diagram" is not correct—could use "plot" or "figure". Perhaps "The plot shows data for potential temperatures above 320 K to focus on" |
| P7 L18 | "include also" → "also include" |
| P9 L10 | "where the vortex" → "when the vortex" |
| P9 L12-13 | the HCl is not really "depleted" wrt to December values, but "HCl makes up a smaller fraction of the total Cly" |
| P9 L30 | the warming "made" higher theta "levels" accessible |
| P10 L13-14 | delete "first"—"model, the model results" |
| P10 L25-26 | "on higher isentropes" → "at higher potential temperatures" |
| P10 L28 | "The modeled vertical shift from HCl to ClONO2 as the dominant reservoir species during repartitioning in March between $360 - 400$ K" |
| P11 L1 | "The phases"—what phases? "The campaign phases"? |
| P11 L19 | "this demonstrates" → "which demonstrates" |
| P11 L27 | comma after HCl "recovery of HCl, as well as" |
| P12 L16 | "achievable cooling" perhaps would be better "cooling that occurs" or "resultant cooling that occurs" |
| P12 L20 | "The same instrumental configuration"? Perhaps "The same instruments used in the present study were previously used for measurements in the Antarctic polar vortex" |
| P12 L21 | "were" → "was", perhaps "was observed." |
| P12 L 24 | "on" → "in" |
| P12 L24 | "also close to" is awkward and vague—"near"? |
| P12 L31 | "showe" → "show" |
| P13 L2 | in standard usage, "supposed to" has a different connotation than the root for "supposition"; here it might be better to say " |

---

## Author Comment (AC1) · 2 Aug 2019

*Our responses to the reviewers' comments are written in italic font. The revised manuscript is attached below and changes are highlighted.*

**Response to anonymous referee #1 (acp-2019-370-RC1)**

The paper by Marsing et al. on chlorine partitioning is clearly within the scope of ACP. The paper is well structured and written, and I do not have much to criticize with respect to its scientific content. Existing work in this research field is adequately referenced. I suggest publication in ACP after consideration of the few minor comments listed below.
*We thank the anonymous referee #1 for his/her positive assessment of the manuscript and for the helpful comments.*

p3 l6 Is the term "chemical gradient" established technical language or slang? I would prefer something like "concentration gradients"
*Although we assume "chemical gradient" to be common technical language, we agree that "concentration gradient" is more specific and, in this case, better points to the issue that we want to address. We changed the term to "concentration gradient".*

p5 l10-12 I suggest to expand somewhat on error estimation. How are the uncertainties estimated? From a statistical analysis of the data or by propagating known ingoing uncertainties through the system? If errors come from a statistical analysis, how are the systematic components estimated? Which error sources have been considered? etc.
*We thank the reviewer for this comment. To be more precise now, we clearly indicate the statistical uncertainties and briefly explain the considered systematic errors. In addition, more detailed information can be found in the given references.*

p5 l11 The term "accuracy" is notoriously ambiguous. Often it is understood as an estimated of the combined random and systematic error, but equally often it is understood as the systematic part alone. One can even find official documents (GUM, ISO and others) which use contradicting definitions of accuracy. I suggest to avoid this term and to use "estimated total uncertainty" or "estimated systematic uncertainty" instead.
*We agree with the reviewer that the usage of the term "accuracy" is not always consistent. In the literature of atmospheric measurements that this manuscript refers to, at least, we found "accuracy" to mean "estimated systematic uncertainty". For this reason, we would like to keep the term in the text, while we attempt to clarify its meaning in the context (see changes relating to the previous comment).*

p6 l21 Using model output from 12:00 UTC only and interpolate data to other locations and times could cause problems in the case of a diurnal cycle. At the altitudes of interest, the diurnal cycle of ClONO2 is not very large but since this is a key condition for the applicability of this method, this assumption should be explicitly stated.
*There could be a misunderstanding here. We do not use the model output a 12:00 UTC in our analysis. Instead, this output (interpolated at the back trajectory locations) serves as an input to another forward run of the CLaMS chemistry module until the observation time. In this way we also respect diurnal cycles. We had hoped that this would become clear from our description. To clarify a little on this, we rewrote a half sentence.*

**Response to anonymous referee #2 (acp-2019-370-RC2)**

The manuscript by Marsing et al. describing the variation in the partitioning of chlorine in the Arctic polar vortex over the course of the winter 2015/2016 is highly topical for Atmospheric Chemistry and Physics. Overall the paper is clear and well-written. The arguments the authors make in the paper are reasonably well supported by the data they present and that data presentation is generally clear and understandable, in particular I appreciated Figure 7. I recommend the manuscript be accepted for publication, but would request the authors consider and possibly address the points below.
*We thank the anonymous referee #2 for his/her positive assessment of the manuscript and for the helpful comments.*

In section 2.2.2, it would be good for the authors to include the correlation value they have used to infer Cly from CFC-12, i.e. Cly = ... CF-12 so that there is a record of what values/relationship was used.

*We thank the reviewer for this comment and agree that it is worthwhile writing down the function at this place. We added the correlation function accordingly.*

Throughout the manuscript, I find the use of "recovery" to describe the repartitioning of Cl into reservoir species somewhat odd. I understand the usage, but it carries a viewpoint of what "normal" is, while indeed the whole repartitioning cycle between active and reservoir species is normal given the conditions. It seems like the discussion could be of "partitioning" or "repartitioning" into the reservoir species. A couple suggestions along these lines are included in the specific comments below.

*We thank the reviewer for this advise. Recovery refers in part to the fact that the reservoir species represent the thermodynamically stable compounds of inorganic chlorine in cloud-free conditions. However, in order to weaken the "normal" viewpoint a little bit, we replaced "recovery" in some places in the text by other formulations including "partitioning" and "repartitioning", and explained the meaning at the first occurrence.*

Figure 4 does not seem particularly important to the argument and is only addressed as an example in a single paragraph. I would think that it could be omitted without effect on the paper.

*Figure 4 shows how shape of the polar vortex (maximum PV) changes from compact and pole-centered at higher altitudes to more filamented and broadly distributed at flight altitude. This should demonstrate how the aircraft measurements could encounter vortex air masses intermittently and even at lower latitudes. Therefore we would like to keep the figure in the paper.*

Is there a reason for plotting the data in Figure 5 as stalactites (inverted y-axis) instead of the standard way with bars increasing upward?

*We thank the reviewer for this comment and redrew the figure with a normal y-axis.*

The caption for Figure 8 indicates there should be grey areas to indicate the model HCl overestimation period, but the figure in my copy does not have any shaded areas.

*This is well observed. We had had some grey areas in an earlier draft of the manuscript. As we removed them from the figure, we did not adapt the caption by mistake.*

Specific minor comments and suggestions for the text:

P1 L1 "process" seems somewhat generic – maybe use "destroy" or similar
*"process" changed to "deplete"*

P1 L6-7 "an altitude dependent shift in the pathway of..." is a little unclear, perhaps something like "an altitude dependence of the pathway for chlorine deactivation with HCl dominating below the 380 K isentropic surface and ClONO2 becoming significant"
*We thank the reviewer for this suggestion and adapted the sentence.*

P1 L20 "abbreviated with" → "abbreviated as"
*changed*

P2 L1 "Besides chlorine, bromine compounds are also important contributors to catalytic ozone destruction."
*included*

P3 L32-33 suggest adding "campaign" – "first major campaign phase"
*included*

P5 L3 "the instrument is already taken into account" could be "the instrument is already accounted for"
*changed*

P5 L10-11 the use "parts per trillion (pptv) of molar mixing ratio" is awkward. The "v" is used for "by volume" which is typically taken as equivalent to by mole. Options would be to delete "of" or replace with "by mole" or volume.
*We agree that this combination is awkward and chose to delete "of".*

P6 L4 "is should be partitioned" – need to use one or the other, "is" or "should be" or perhaps "is expected to be"
*changed*

P6 L20 "winder" → "winter"
*changed*

P7 L1 "of N2O ... measurements" → "measurements of N2O and potential temperature" (although potential temp is calculated, not measured)
*changed*

P7 L1    delete "of" from "of PV", and delete "also"
*changed*

P7 L3    "diagram" is not correct – could use "plot" or "figure". Perhaps "The plot shows data for potential temperatures above 320 K to focus on"
*changed*

P7 L18    "include also" → "also include"
*changed*

P9 L10    "where the vortex" → "when the vortex"
*changed*

P9 L12-13    the HCl is not really "depleted" wrt to December values, but "HCl makes up a smaller fraction of the total Cly"
*We thank the reviewer for the comment; this wording seems indeed more appropriate.*

P9 L30    the warming "made" higher theta "levels" accessible
*changed*

P10 L13-14    delete "first" – "model, the model results"
*changed*

P10 L25-26    "on higher isentropes" → "at higher potential temperatures"
*changed*

P10 L28    "The modeled vertical shift from HCl to ClONO2 as the dominant reservoir species during repartitioning in March between 360 – 400 K"
*changed*

P11 L1    "The phases" – what phases? "The campaign phases"?
*"Phases" replaced by "periods".*

P11 L19    "this demonstrates" → "which demonstrates"
*changed*

P11 L27    comma after HCl "recovery of HCl, as well as"
*included*

P12 L16    "achievable cooling" perhaps would be better "cooling that occurs" or "resultant cooling that occurs"
*changed*

P12 L20    "The same instrumental configuration"? Perhaps "The same instruments used in the present study were previously used for measurements in the Antarctic polar vortex"
*changed*

P12 L21    "were" → "was", perhaps "was observed."
*changed*

P12 L 24    "on" → "in"
*changed*

P12 L24    "also close to" is awkward and vague – "near"?
*Changed to "in the vicinity of"*

P12 L31    "showe" → "show"
*changed*

P13 L2    in standard usage, "supposed to" has a different connotation than the root for "supposition"; here it might be better to say " [sic]
*We are sorry to note that the last comment by the reviewer was clipped. But we get the point, and replaced "supposed" by "presumed".*

[revised manuscript text omitted]